# Impact of Leadership Behavior on Entrepreneurship in State-Owned Enterprises: Evidence from Civil Servant Management Aimed at Improving Accountability

**Dao T. T. Thuy [1], Truong Quoc Viet [2], Vu Van Phuc [3], Thi-Hong-Diep Pham [4], Nguyen Thi Ngoc Lan [1] and Huong Ho [1,\*]**

[1] Faculty of Politics, Vietnam Youth Academy, Hanoi 100000, Vietnam
[2] Faculty of Administration, Hanoi University of Internal Affairs, Hanoi 100000, Vietnam
[3] Scientific Council of Central Party Organizations, Hanoi 100000, Vietnam
[4] Faculty of Political Economy, VNU University of Economics and Business, Hanoi 100000, Vietnam
[\*] Correspondence: hohuong112007@gmail.com

**Abstract:** This paper systematizes the theoretical foundation and empirical evidence of the impact of leadership behavior on entrepreneurship in state-owned enterprises, and the difference between state-owned enterprises that applied management based on the accountability of leaders, and those that did not. The paper uses the OLS regression model to identify the impact of leadership behavior on the entrepreneurship of state-owned enterprises by using data from a survey of 259 civil servants in Vietnamese state-owned enterprises. In our sample, 109 respondents belonged to the category of state-owned enterprises that did not apply management based on accountability, and 140 were state-owned enterprises that applied management based on accountability. The findings show that leadership behavior has a positive impact on the entrepreneurship of state-owned enterprises that do and do not apply management based on accountability, with results of 0.305 and 0.022, respectively. Moreover, the regression model is used to identify the factors that influence leadership behavior, including vision and wage policy having a statistical significance and positive impact on the leadership behaviors in all the state-owned enterprises. Additionally, some factors, including policy building, hi-tech usage, culture, teamwork, and training policy have positive impacts on leadership behavior in Model 1, and encouragement, monitoring, responsibility, and recruitment policy had positive impacts in Model 2 among the state enterprises. Additionally, this paper recommends some policies to promote leadership behavior in state enterprises in Vietnam.

**Keywords:** leadership behavior; entrepreneurship; Vietnamese state-owned enterprises; civil servants; accountability



## 1. Introduction

State-owned enterprises have an essential role in modern economies, and are considered a vehicle for accelerating economic growth. Moreover, these enterprises also provide public services and public values. State-owned enterprises are internationally defined as "any corporate entity recognized by national law as an enterprise, and the state exercises ownership" (OECD 2018).

State-owned enterprises are partially or wholly owned by the state, and are established to facilitate growing co-operation between the private and public sectors as a result of structural transformations in the economy (Larsen et al. 2021). State-owned enterprises help to change markets, improve effectiveness and efficiency, and provide better services, making a clear division of responsibility between owner (ministry) and management (Statskonsult 1998). A state-owned enterprise can take different forms and pursue a wide range of activities. In Vietnam, state-owned enterprises are a part of the state economy, and have always been identified as playing a key role in the national economy. In order to

promote the performance of state agencies, including state-owned enterprises, Vietnam has issued the Master Program on State Administrative Reform for the period from 2021 to 2030. One of the focuses of the program is to build a contingent of professional, capable and qualified cadres and civil servants to meet the requirements for the development of the country, focusing on the increased accountability of leaders.

The concept of public accountability specifically points to the belief that the public has a "right to know" or the right to obtain reliable information from the government. It is common knowledge that this concept of accountability exists when there is one party responsible for reporting decisions and actions to another party. Thus, accountability will exist if the agent receives resources and responsibilities from the principal (Keay 2017). Accountability is a requisite for social order and one of the cornerstones of good governance. The literature shows that accountability is necessary for the effective function of organizations. Accountability has a very broad and interpretable meaning, depending on the relevant circumstances and contexts. Accountability can be interpreted as an inherent relationship between the parties who give and receive an operation (Roberts and Scapens 1985). The concept of accountability is also closely related to power, where power can be delegated, but responsibility cannot be relinquished.

Leadership means the process of influencing people to achieve an organization's goals and guiding others' behavior to obtain the stated objectives (Mehmet and Chowdhury 2020). Leadership arranges for human resources and materials to be used effectively in a particular way. Thus, leadership plays an important role in the allocation of resources and expansion of business, and is a necessary factor for the development of enterprises. According to Simon (2017), leadership behavior can encourage employees to develop and improve their abilities and skills optimally by creating suitable conditions. Leader behavior has been shown to improve entrepreneurship; a well-managed business has an increased likelihood of reaching its corporate objectives and enhancing its overall entrepreneurship, focusing on specific entrepreneurial behavior or exceptional abilities, such as recognizing and exploiting entrepreneurial opportunities (Renko et al. 2015).

Leaders' behaviors can play a necessary role in improving entrepreneurship. Their behavior can affect an organization not only in terms of success and efficiency, but also in terms of how effectively state-owned enterprises deliver services to their customers in a funding-constrained environment (Zerbinati and Souitaris 2005). Leaders are also an important source for acquiring resources, changing strategies based on knowledge of the changing environment (Covin et al. 2019), and motivating employees to be entrepreneurial through incentives and creating an entrepreneurial culture (Kim 2010). Therefore, enhancing entrepreneurship is very important to maintaining a business effectively. Leadership behavior with accountability improves the role of leaders in managing state-owned enterprises.

The objective of this study is to determine the impact of leadership behavior in accountable management on entrepreneurship in the Vietnamese state-owned enterprises in the Master Program on State Administrative Reform for the period from 2021 to 2030 period. To do so, this study adopts the following research objectives:

- It aims to evaluate the relationship between leadership behavior and entrepreneurship in state-owned enterprises and compare the difference in this impact between the state-owned enterprises that do and do not apply management with accountability. From our findings, we recommend that the state enterprises applying management with accountability are affected in their entrepreneurship more than the state enterprises not applying management with accountability.
- It aims to identify factors that affect their leadership behavior in state enterprises. In particular, by using models for the two types of state enterprises, the study also analyzes and evaluates the different factors for each the state enterprise.

## 2. Literature Review

Many empirical studies have reported a positive relationship between leadership behavior and entrepreneurship. Leadership success is likely to be influenced significantly

by context variables, such as firm development or company structure. Accordingly, the context in which leadership takes place should be incorporated into research. Anjeza (2017) shows that leadership and organization effectiveness communication is essential, especially for the internal functioning of the organization, and it realizes the integration of all managerial functions. Communication helps to set targets for each department, organizes human resources and all other resources effectively, and helps employees better understand the decision-making process. The role of the leader in the organization is a decisive role since any action taken by him affects all work processes throughout the organization. Idris and Ali (2008) conclude that leaders should have the capability to materialize a formulated vision through managing all quality components to shift the firm toward quality managerial practices.

An exploratory study based on Qatar Steel Company performed by Ismail (2009) finds implementation difficulties of different quality programs due to lacking of management support. Such finding is absolutely opposite of Salahedin's notion that management support and commitment lead to an organizational climate of cooperation, which produce positive results like productivity increases, quality improvement and enriched management styles. Pera (2019) studied a management role of giving flawless direction to the employees, while Pannirselvam and Ferguson (2001) revealed the strength of relationship between organizational performance and quality management where they found leadership as a considerable element which indirectly or directly influencing all system units. Based on the Malaysian firms, Idris and Ali (2008) conducted a study showing adaptive capacity as a key for the firms to survive in emerging global economic order. The authors have termed these means to the effective managerial approaches for organizational change and leaders create the vision through efficient communication and motivate employees to achieve such vision.

An organization creates its own dynamics and culture to overcome the critical issues to achieve excellence. Through the behavior and practices, both informally and formally, leaders instill the value to create a unique organizational culture. The critical role of leadership regarding system philosophy which organization must adopt in a purposeful manner to ensure excellence (Bass 1990). Ostrom (2005) confirms that managers in public organizations play an important role because they represent a particular form of leadership focused primarily on problem-solving and putting heterogeneous processes together in complementary and effective ways and they set an organization's strategic goals and making day-to-day decisions along with communicating these goals to employees (Bruneel et al. 2010).

Yukl (2006) thinks that leadership behavior is associated with adapting to change in the environment; increasing flexibility and innovation; making major changes in processes, products, or services while Amagoh (2009) shows that leadership competencies and effectiveness several scholars have studied leadership competencies as potential factors influencing leadership effectiveness. Leadership competencies are considered a promising predictor of a leader's performance and the organization's functionality. Employees who follow and evaluate their leader's competencies positively, perform their tasks better, feel a high level of job satisfaction, are more motivated, and actively support cooperation and communication between all participants in the organization (Mayoral and Vallelado 2015).

Leadership researchers claim that effective leadership may also include a strong emotional, ethical and cultural component which great leaders usually use to mobilize followers (Koman and Wolff 2008). Effective leaders have mobilization competencies which they use to overcome emotional, ethical and cultural constraints. Thus, they encourage followers to sacrifice and achieve great goals (Miao et al. 2018). Moreover, to be effective in mobilizing others, leaders need to handle informational constraints related to leadership competencies and effectiveness their employees' personal needs and ambitions (Bass 1990). Furthermore, leaders who are able to mobilize their employees deal well with power plays (linked to political constraints) and do not need formal rules (entitlement constraints) (Noe et al. 2017).

To generate entrepreneurial activity, leaders need to be able to motivate, design tasks, and delegate and coordinate human resources (Currie et al. 2008). They are also able to effectively manage setting goals, allocating labor, and enforcing sanctions. They initiate the structure for their followers, define the roles of others, explain what to do and why, establish well-defined patterns of organization and channels of communication, and determine the ways to accomplish assignments (Bass 1990). To increase the absorptive capacity of an organization, the leaders can help improve the skills and the entrepreneurial behavior of employees such that employees can recognize the value of new opportunities and apply their knowledge to increase entrepreneurial activity. According to Schwarzmüller et al. (2018), the common problems generated by the digitalization of organizations are worker alienation, weak social bonding, and poor accountability. It is therefore extremely important that the leaders support and help followers in dealing with the challenges of greater autonomy and increased job demands, by adopting coaching behaviors to promote their development, provide resources, and assist them in handling tasks.

Furthermore, a leader's vision competencies translate into the ability to create future visions for oneself and one's followers, making them sufficiently distant and attractive to mobilize followers to act. Bandura (1997) offers a comprehensive framework to explain the impact of the leaders on the followers' task-related performance. This theory proposes that leadership exerts its influence on followers' knowledge in large. In addition, leaders may overcome cultural constraints. The leaders and their followers need to speak the same language. It might help deal with motivational, informational. Thus, value-creation competence allows leaders to find similar vision to their employees, overcome constraints and ensure leaders' effectiveness at the team and organization levels. Simon (2017) shows that leadership is focused on exchange relationships between leaders and their followers. In this exchange process, leaders appeal to the self-interest of their followers as a means of motivating them towards the achievement of specific tasks. Additionally, the role of leaders to ensure the performance excellence of an organization is immense. Leaders play a pivotal role in motivating the employees to guide their behavior to the right direction to generate the desired output. They also extend that successful and effective leadership means fundamentally influencing others by establishing a direction for collective effort and managing, shaping, and developing the collective activities according to this direction (Cogliser and Brigham 2004).

According to Ahmed et al. (2010), leaders play a vital role to boost it up an increase productivity leads an organization to achieve higher efficiency and hence, effectively achieving the strategic objectives of the organization and are required to ensure that the proper protocols are followed and they must evaluate the performance of employees against clearly defined expectations. In addition, leaders must provide their employees with immediate assistance when requested. Therefore, communication must be permitted to flow freely, absent of interruption. Consideration leadership style involves subordinates in the empowerment process, supports subordinates to think and express ideas, and treats subordinates fairly through good judgment. It is in line with the interactive control system. The interactive control system consists of a formal information system for managers who use and involve themselves regularly and personally to make decisions about subordinates' activities, intending to stimulate new ideas and strategies and provide solutions in overcoming problems (Efrizal 2012). Leaders' behavior can help with developing both team cohesion by sharing the belief in a group's collective capability to organize and execute courses of action required to produce given levels of goal attainment (Kozlowski and Ilgen 2006). Leaders operating within the set of specific norms that are deeply rooted in the organization experience limited effectiveness, and they can influence their team only when they act in line with these norms.

Moreover, Savage and Sales (2008) argued that leaders endowed with this competence understand the dynamics in the organization's environment, which is often concerned with cultural constraints, discern patterns and trends in various industries linked to overcoming information constraints, and can predict interactions among various forces that help to

handle political constraints. Anticipative leaders might be very influential, especially at the stakeholder level. Consideration leadership style focuses on promoting subordinates through welfare support and comfortable relationships. Consideration leadership style has also been proven to build a working atmosphere of mutual trust with subordinates, respect the ideas put forward by subordinates, and consider subordinates' feelings. Based on the above statement, evaluation fairness will be created. According to Dao (2021), smaller organizations represent a simpler and more integrated social system, with fewer people, fewer levels of organizational hierarchy and less subdivision of work. To compensate for some of the problems of increasing size, larger organizations, leadership may make more use of formal structures, systems and procedures. Similarly, leader can create a positive organizational environment that fosters a strong sense of collaboration and unity among employees has become vital for leaders to have. They need to integrate these social skills with the ability to master a variety of virtual communication methods (Damti and Hochman 2022).

Besides, related to the relationship of accountability in the state-owed sector, some studies show that the relationship has indirectly regarded the public as one of the users of information and services offered by the government (Lynn and Stein 2001). Accordingly, the public has the right to know the extent to which these financial resources are used and managed by the government (van der Voet 2016). Therefore, the government should be more open to disclosing information needed by the public (Steccolini 2004). This is because it is a responsibility that needs to be implemented, in addition to acting as one of the ways to implement their accountability to the role of government to the people. Accountability has a very broad and interpretable meaning according to the relevant circumstances and contexts. Traditionally, accountability can be interpreted as an inherent relationship between the parties giving and receiving an operation (Roberts and Scapens 1985). The concept of accountability is also closely related to power where power can be delegated but responsibility cannot be relinquished. This means that manager must be held accountable for the actions taken by subordinate employees. The leadership practices helps to achieve the quality and positive outcomes. To endorse leadership as described requires personal and managerial authority being used in an appropriate balance (Greenfield 2007).

Greenfield (2007) confirms that effective leadership can drive improvements in teamwork, quality and safety, and innovation. The leaders of an organization play an important role in creating an organizational culture of innovation and understanding leadership behavior is critical for the organizations and the stakeholders. Motivation remains one of the major challenges that corporations face today, especially when it must be combined with the efficiency and effectiveness of the organization (Mehmet and Chowdhury 2020). With this current development the state enterprises require not only the manager but also a manager with leadership charisma. Therefore, when the behavior of the leader is too different from the expectations of the followers, undesirable consequences can happen and weaken individual and work group performance. Gonzalez and Firestone (2013) found that leaders play a key role by interpreting state and federal policies in ways that influence local interpretation. There are possibilities of a gap in the leadership style posed by the leader that could affect the accountability. The reputations of leaders have impact on the degree of formal accountability mechanisms for their work-related decisions and actions (Bryant 2010). This highlights the complex relationships that appear among leader reputation, trust, and accountability which also can facilitate leader performance and effectiveness. In order to achieve greater accountability within the public sector organization, focus on developing the appropriate characteristic of leadership must be achieved.

There are some models used to evaluate impact of leadership behavior on entrepreneurship such as ordinary least squares, ordinal logit model, multiple regression models, hierarchical moderated regression, and SEM-PLS. Mehmet and Chowdhury (2020) use the ordinary least squares (OLS) and ordinal logit model (OLM) analysis to demonstrate that relations-oriented leadership is an effective leadership behavior that may lead to higher entrepreneurship among employees, employee job level affects how leadership types influence public sector entrepreneurship by some control variables including size, location, job

level, education, gender, types. Simon (2017) provides the results of hierarchical moderated regression analyses for the performance of the enterprises. This article explores the role of the leadership and outlines a model of leadership behavior of leaders influencing by using a sample of 102 enterprises and 372 employees. The results indicate that leadership has a significant and positive effect on the performance of the enterprises.

Regarding the influence between the considerations of leadership style on evaluation justice, Efrizal (2012) uses SEM-PLS, a multivariate analysis that can test measurement models and structural models. The results of this study also indicate an indirect effect between the relationship between the leadership style of the consideration and the fairness of evaluation mediated by objective diagnostic, objective interactive, subjective diagnostic, and subjective interactive with the form of partial mediation. In addition, Valaskova et al. (2020) has applied bankruptcy models examined in the agricultural section in Slovak economy. The authors have tested the financial stability from household farms and proposed some recommendations to mitigate the hidden financial risk and improve the management method.

Overall, many empirical studies have reported a positive relationship between leadership behavior and entrepreneurship. The role of the leader in the organization is a decisive part since any action taken by him affects all work processes throughout the organization. Through the behavior and practices, both informally and formally, leaders instill the value to create a unique organizational culture. In addition, diverse determinants of leadership behavior highlighted in the literature include manager skills (vision, encourage, policy building, monitoring, professional knowledge (knowledge, hi-tech usage, responsibility, communication), environment variables (work culture, teamwork, wage policy, recruitment policy, training policy, etc.). Moreover, some studies have used the linear regressive model to examine the impact of leadership behavior to entrepreneurship, effectiveness, employee innovative, etc. (linear regressive, binomial logarit regression, multinomial logit model, SEM-PLS, etc.). Although some of those studies did examine the relation between entrepreneurship and leadership behavior in the enterprises, not even one emphasized impact of leadership behavior on entrepreneurship of state-owned enterprises: Evidence from civil servants' management according to improving accountability

## 3. Materials and Methods

In Vietnam, in order to promote the performance of state agencies, including state-owned enterprises, Vietnam has issued the Master Program on State Administrative Reform for the period from 2021 to 2030 period. One of the focuses of the program is to build a contingent of professional, capable, qualified cadres and civil servants to meet the requirements and the development of the country, in which, focusing on the increase accountability of leaders. Our main information for the analysis was got from a survey of civil servants in state enterprises in Vietnam applying issued the Master Program on State Administrative Reform. We select these state enterprises in Ha Noi, Ho Chi Minh city, Lai Chau. The regions were purposely chosen because the state-owned enterprises in these regions are ones of the most important state-owned enterprises in Vietnam with large state-owned economic groups, 100% capital of the corporation owned by the state, over 50% capital of the corporation owned by the state.

To get a sample, we drew a random sample of civil servants from a complete list in each enterprise, and we then selected civil servants from among the listed members. To assure the sample representation of survey data, at each studied enterprise, we select. We select interviewees from those lists supplied by local authorities. A semi-structured and structured questionnaire was used to collect data from civil servants in these state enterprises. Face-to-face interviews were conducted with 259 civil servants whose job level of these civil servants are APS level, executive level, senior executive service (109 civil servants in the enterprises not applying accountability and 150 civil servants in the enterprises not applying accountability) to collect information on leadership behavior, entrepreneurship, effectiveness of enterprises.

To estimate the role of leadership behavior for the entrepreneurship in public organizations, Mehmet and Chowdhury (2020) use the ordinal logit regression model (OLS) with comparing the results of the models with variables related behaviors of leaders. Ordinary least squares (OLS) regression is a statistical method of analysis that estimates the relationship between one or more independent variables and a dependent variable; the method estimates the relationship by minimizing the sum of the squares in the difference between the observed and predicted values of the dependent variable configured as a straight line. In this study, we use the OLS with these variables to examine the impact of leadership behavior on the entrepreneurship in Vietnamese state-owned enterprises including labor scale, location, tenure, education, gender, type and the specific features of the state-owned enterprises are affected in the new stage in Vietnam with applying the Master Program on State Administrative Reform for the period from 2021 to 2030 period including hi-tech usage, manufacturing sector, organizational model (Dao 2021). The result of the OLS regression method to show the factors that influence entrepreneurship in the Vietnamese state-owned enterprises. However, some unobserved factors also influence the entrepreneurship. If this occurs, the result of the OLS model can generate a biased parameter as estimating the simple regression analysis of the entrepreneurship based on dichotomous variable associated with leadership behavior will overvalue how much the leadership behavior will affect the entrepreneurship. Therefore, the model used to identify factors that influence the entrepreneurship in the Vietnamese state-owned enterprises:

$$P(1,0) = \alpha Z_i + \varepsilon_i \tag{1}$$

where $\alpha$ is constant, P is a dummy variable (1 for applying accountability and 0 for not applying accountability), $Z_i$ is a set of respective observed factors expected to influence decision to applying accountability in the enterprises, $\varepsilon_i$ is a set of respective observed factors expected to influence the entrepreneurship in Vietnamese state-owned enterprises.

In addition, the paper uses the regression method to show the factors that influence the behavior of leaders in Vietnamese state-owned enterprises. The first group of variables is vision, encourage, policy building, monitoring. We use these variables in the analysis for the manager's skills. The second group includes variables on professional knowledge, e.g., knowledge, hi-tech usage, responsibility, communication. The third group includes the work environment variables, i.e., work culture, teamwork, wage policy, recruitment policy, training policy (see Table 1).

**Table 1.** Definition of explanatory variables used in the regression model.

| Variables | Definition | Mean/Share | Std. Dev |
|---|---|---|---|
| Entrepreneurship | - Give ideas proactively to improve the way you work <br> - Willing to work overtime when required <br> - Complete all tasks well to help the enterprises achieve its goals | 2.63 | 1.24 |
| Leadership behavior | - Have high professional competence <br> - Good communication skills <br> - Good orientation and planning ability <br> - Good supervisory capacity <br> - Treat employees fairly <br> - Listen to everyone's opinion <br> - Innovation at work <br> - Effective management | 3.21 | 1.15 |

**Table 1.** *Cont.*

| Variables | Definition | Mean/Share | Std. Dev |
|---|---|---|---|
| Startup year | Number of years the company was established | 3.05 | 1.21 |
| Tenure | The period of time when someone holds a manager role | 3.03 | 1.18 |
| Vision | The ability to concentrate on the most important aspects of self or business | 3.48 | 1.23 |
| Encourage | Focuse on the individual's strength and contributions in order to drive their motivation and performance to a higher level | 3.38 | 1.20 |
| Policy building | Give new policies for the enterprises | 3.35 | 1.23 |
| Monitoring | Ensuring that work goals are being met, that ethical standards are upheld, and that relevant financial and fiduciary duties are fulfilled | 3.47 | 1.23 |
| Knowledge | Manager's understanding | 3.39 | 1.26 |
| Responsibility | Including decision-making, coaching, mentoring, developing the team's skills and managing conflict | 3.32 | 1.31 |
| Communication | Gross farm revenue (VND/ha) | 3.34 | 1.18 |
| Work culture | Collection of attitudes, beliefs, and behaviors that make up the regular atmosphere in a work environment. | 3.28 | 1.20 |
| Teamwork | The ability to work in groups | 3.49 | 1.20 |
| Wage policy | Preferential salary for leader | 3.48 | 1.23 |
| Recruitment policy | Preferential recruitment for leader | 3.31 | 1.23 |
| Training policy | Preferential training for leader | 3.45 | 1.01 |
| Manufacturing sector | 0 = agriculture<br>1 = Industry<br>2 = Service | 12.5%<br>57.9%<br>42.1% | |
| Scale | The scale of labor<br>Small (less than 100 employees)<br>Medium (100–500)<br>Large (over 500) | 49.7%<br>27.7%<br>22.6% | |
| Location | Rural<br>Urban | 26.3%<br>73.7 | |
| Organizational model | 50% state-owned capital<br>100% state-owned capital | 56.3%<br>43.7% | |
| Gender (male) | Male<br>Female | 58.8%<br>41.2% | |
| Hi-tech usage | Ability for using hi-tech | 34.7% | |

Source: author's survey (2022).

## 4. Results and Discussion

### 4.1. Impact of Leadership Behaviors to Entrepreneurship in the State-Owned Enterprises

The survey also provides some background information on state enterprises. It shows that the scale of labor of these enterprises is medium and their manufacturing sector concentrates in industry and service. These 50% state-owned capital occupy more than a half with 56.3%. These state enterprises applying manager according to accountability occupied 94% while the other states enterprises are 0.6%. According to the survey with 259 state enterprises, the results show that the average revenue of these enterprises per year is 166.366 billion VND. There is a high distance in the revenue between the enterprises, the biggest enterprises are in industrial manufacturing sector. The average of state-owned states applying manager according to accountability is 2.936 billion VND while this figure of state-owned states not applying manager according to accountability is 2.472 billion VND.

Table 2 presents the results of the OLS model on effects of leadership behavior to entrepreneurship of state-owned enterprises. The variables, which are age, education,

location, scale, field, job level, tenure are meaningful in explaining the impact of leadership behavior to the entrepreneurship of the state-owned enterprises. In Table 3, test F of regressive is significant ($p < 0.05$), which shows that there is no error in selection.

**Table 2.** Effects of leadership behavior to the entrepreneurship of the state-owned enterprises.

| Variables | Model 1 Applying Accountability | | Model 2 Not Applying Accountability | |
|---|---|---|---|---|
| | Coefficient | SE | Coefficient | SE |
| Leadership behavior | 0.022 | 0.171 ** | 0.305 | 0.049 *** |
| Age | −0.344 | 0.169 *** | 0.147 | 0.017 ** |
| Education | −0.332 | 0.215 * | −0.025 | 0.020 * |
| Location | 0.321 | 0.241 ** | 0.030 | 0.022 *** |
| Labor scale | 0.525 | 0.171 ** | 0.007 | 0.145 ** |
| Manufacturing sector | 0.140 | 0.105 ** | −0.002 | 0.015 *** |
| Job level | −0.068 | 0.117 ** | 0.259 | 0.010 ** |
| Organization model | 0.061 | 0.213 *** | 0.524 | 0.221 *** |
| Hi-tech usage | 0.041 | 0.214 ** | 0.057 | 0.142 ** |
| Tenure | 0.030 | 0.321 ** | 0.152 | 0.126 ** |
| Constant | 1.877 | 0.503 | 0.670 | 0.070 |

Note: Robust standard errors in parentheses. *** $p < 0.01$, ** $p < 0.05$, * $p < 0.1$. Source: author's survey (2022).

**Table 3.** Effect of the factors on enhancing leaders of the state enterprises.

| Variables | Model 1 Applying Accountability | | Model 2 Not Applying Accountability | |
|---|---|---|---|---|
| | Coefficient | Std. Err | Coefficient | Std. Err |
| Management skills | | | | |
| Vision | 0.058 | 0.235 *** | 0.136 | 0.735 *** |
| Encourage | −0.244 | 0.282 ** | 0.070 | 0.732 ** |
| Building Policy | 0.157 | 0.344 *** | −0.123 | 0.749 ** |
| Monitoring | −0.155 | 0.362 ** | 0.058 | 0.087 ** |
| Professional knowledge | | | | |
| Knowledge | −0.049 | 0.286 ** | −0.098 | 0.067 * |
| Hi-tech usage | 0.362 | 0.405 ** | −0.003 | 0.059 ** |
| Responsibility | −0.163 | 0.386 ** | 0.035 | 0.072 ** |
| Communication | −0.667 | 0.311 * | −0.037 | 0.063 ** |
| Work environment | | | | |
| Culture | 0.050 | 0.361 ** | −0.057 | 0.043 * |
| Teamwork | 0.241 | 0.215 ** | −0.034 | 0.739 ** |
| Wage policy | 0.252 | 0.365 *** | 0.051 | 0.665 *** |
| Recruitment policy | −0.523 | 0.274 ** | 0.147 | 0.775 ** |
| Training policy | 0.668 | 0.260 *** | −0.971 | 0.082 ** |
| Number of observations | 109 | | 150 | |
| R-squared | 0.401 | | 0.214 | |

Note: Robust standard errors in parentheses. *** $p < 0.01$, ** $p < 0.05$, * $p < 0.1$. Source: Author's survey (2022).

The finding demonstrates that leadership behavior has effect on the entrepreneurship in the state enterprises. However, in the enterprises applying management according to accountability, the impact of leadership behavior is less than other enterprises with 0.022 times in model 1 and 0.305 in model 2. Previous studies have echoed our conclusion on the role of leadership behavior in the public organizations. Mehmet and Chowdhury (2020) show that all types of leadership behavior are positively associated with public sector entrepreneurship and the effect is larger for relations-oriented leadership, followed by the change-oriented leadership. Effective leaders can create an entrepreneurial climate through empowering employees, provide them with incentives, improve employees' capabilities and motivations, build team cohesion, help them to face with a challenging sit-

uation or improve assigned tasks which is helpful for better performance and productivity (Collins and Smith 2006).

In the model 1, the state—owned enterprises do not apply the method of management according to improving accountability. Labor scale, location, manufacturing sector have the highest positive impact on the entrepreneurship in the state enterprise with 0.525, 0.321, 0.140, respectively. However, in the model 2, these variables do not affect as high as in the model 1. Variables age and job level do not have positive impact in the entrepreneurship of the state-enterprises not applying accountability with −0.344, −0.068 but improve the entrepreneurship of the enterprises applying accountability.

In addition, the result demonstrates that the organization model has an influence for the entrepreneurship of all the state enterprises; however, having the highest influence in model 2 with 0.524. The reason is that almost 50% state-own enterprises usually have to apply accountability to continue their effectiveness.

The other factor affecting entrepreneurship is hi-tech usage, with the rate of 0.041, 0.057 units for all state-owned enterprises, and there is a slight difference between two groups. Digital transformation and applying hi-tech in business processes is a necessary and relevant stage in the development of the enterprises. Hi-tech usage is a change of form of operations, restructuring of organizational structure, application of new business models, new sources and forms of income, attracting a wider range of consumers, bringing customer service to a new level, mixing areas of operation in new formats, including in the form of digital platforms. The digital transformation strategy drives changes in business models and uses technology in managing to create the opportunities needed by an enterprise to become a digital business (Pronchakov et al. 2022).

Besides, it is also noted that the entrepreneurship can result from the factor including tenure with 0.030 (in the model 1) and 0.152 (in the model 2). The finding indicates that these factors have a statistical meaning and influence improving entrepreneurship positively. However, education variable is a discrepancy between model 1 and model 2 with −0.332 and −0.025, respectively. In contrast, age would decrease the entrepreneurship of the enterprises not applying accountability (−0.344) but increase the entrepreneurship in the enterprises applying accountability (0.147).

*4.2. Effects of Factors on Leadership Behaviors in the State-Owned Enterprises in Vietnam*

After identifying the determinant leadership behavior on the entrepreneurship of state enterprises, the authors examined the impact of the factors on enhancing ability of leaders in the state-owned enterprises.

Table 3 presents the results of the regression model on the leadership behavior. The variables, which are vision, encourage, policy building, monitoring, knowledge, hi-tech usage, responsibility, communication, culture, teamwork, wage policy, recruitment, training have meaning in explaining the impact of factors on the leadership behaviors in the state-owned enterprises in Vietnam. According to Jaroslav (2013), leaders play a very big role in creation of moral climate and culture in an organization. Leadership practice has a significant impact on employees' performance and thus generates an organizational culture where employees are motivated to contribute productively in the organization. Therefore, leaders' act as a positive ethical model which contributes to better the performance of employee.

In Table 3, test F of regression has a meaning, which shows that there is no error in selection. By using hi-tech in work, leaders can increase their work effectiveness considerably in model 1 but decrease this in model 2. Vision and wage policy can improve their management ability for all state enterprises with 0.058 and 0.136 times. Similarly, Simon (2017) reveals that leaders need to be able to motivate, design tasks, delegate and coordinate human resources. They are also able to effectively manage to set goals, allocating labor, and enforcing sanctions.

The other factor affects to the leadership behavior in all state enterprises is wage policy with the rate of 0.252 and 0.051. Leaders will make effort to improve management skills

when they have high salaries. Koronios et al. (2019) studied among other motivators such as fair wages, promotion incentives, self-esteem and creativity shedding light on their impact on work performance and work performance was significantly explained by pay incentives, self-esteem and corporate ethical values.

Besides, according to the results of model estimation, the increase of a training course for a leader will lead to an increase of 0.668 units of the skill management in the model 1. This can be explained by the efficiency of training will be able to improve the management skill. Leadership is related to the development in giving the direction, support communication, other management skills to bring everyone together and make them perform their tasks in the desired form. Leader is the person who can identify well suitable person and put that person for the task to enhances the effectiveness. This is not possible unless the leader gets the trust and belief of his subordinates. It is a deliberate process of continuous interaction between leader and subordinate to make the task happen with their coordination and knowledge. Effective leaders engage their management skills and translate these into explicit behaviors to positively influence change initiatives (van der Voet 2016; Ahmed et al. 2010; Anjeza 2017).

Similarly, the efficiency of hi-tech usage, culture, teamwork have influenced positively to the management skill of leaders in the state enterprises (0.362, 0.050, 0.241, respectively) in the model 1 and encourage, monitoring, responsibility, recruitment policy (0.070, 0.058, 0.035, 0.147, respectively) in the model 2. However, these factors have positively in the model 1 but do not have positively in the model 2. This proves that applying accountability in management of Vietnamese state enterprises having the effect in improving the leadership behavior.

## 5. Conclusions

In summary, as can be seen from the model's results, leadership behavior plays an essential role in improving the entrepreneurship in the Vietnamese state-owned enterprises. So, the leadership behavior can be considered an important factor in enhancing productive effectiveness in the state-owned enterprises, help officers proactively giving ideas to improve the ways for work, willing to work overtime when required and completing all tasks well to help the enterprises to achieve their goals.

From the research findings, it can be concluded that the factors impact the entrepreneurship in the state-owned enterprises including:

First, the leadership behavior affects positively to the entrepreneurship in all state-owned enterprises including the enterprises applying or not applying management following accountability with 0.305 and 0.022, respectively. Especially, in the state-owned enterprises applying management according to accountability, the effect of behavior of leader is higher than in the other state enterprise. It improves that accountability can enhance the ability of leaders and the entrepreneurship of the state-owned enterprises.

Second, the other positive factors include location, scale, manufacturing sector, organization model, hi-tech usage, tenure in the model of state enterprises not applying management according to accountability and age, location, scale, job level, organization model, hi-tech usage, tenure in the model of state enterprises applying accountability.

Third, to evaluate determinant factors on leadership behavior, the finding shows the vision and wage policy have a statistic significance and positive impact on the leadership behavior in all state-owned enterprises. It concludes that leadership is essential for the success of any organization, business or team as well as leadership need to have a well-defined vision. This helps them to prioritize their goals and keep their duties on track. Additionally, wage policy is also important to encourage the leaders in their work with 0.252 (the model 1) and 0.051 (the model 2).

Besides, some factors including policy building (0.157), hi-tech usage (0.362), culture (0.050), teamwork (0.241), training policy (0.668) have positive impact in leadership behavior in the model 1 of the state enterprises and encourage (0.070), monitoring (0.058), responsibility (0.035), recruitment policy (0.051) in the model 2 of the state enterprises.

As a result, in the next years, there are some solutions need to be created to promote the leadership behavior in the state-owed enterprises:

Implement a number of measures to improve the leadership capacity in the state-owned enterprises, such as applying on-site capacity, building training methods and improving skills to empower subordinates in making decisions which help to increase finding out new methods, directions and approaches to overcome difficulties. Through practical work, new capacities of civil servants will be formed and perfected.

Improve the quality of training and fostering managers. The government needs to combine the form of training and retraining according to rank and grade standards with the form of training and retraining according to job positions. On the basis of standard competency frameworks, the state-owned enterprises build training and retraining programs for their leaders and combine to organize various forms of training suitably, intersperse with training sessions and exchanging management experience.

Enhance the leader's vision. To be a visionary leader, the leader needs to build a clear vision of the future. More importantly, this vision needs to be shared with all members of the organization.

Improve the wage policy in the state enterprise sector according to the market mechanism. The state enterprise is entitled to decide on the salary policy according to the general principle ensuring the harmony between the employee's interests and the employer in the context of the undeveloped labor market in Vietnam.

However, similar to other studies, there are some limitations to ours. Our study mainly focuses on testing the impact of the leadership behavior on entrepreneurship of the state-owned enterprises. In the meantime, this is the second stage of the research project to study the role of leadership behavior on other types of enterprises.

**Author Contributions:** Conceptualization, H.H., D.T.T.T. and N.T.N.L.; methodology, H.H., T.-H.-D.P.; software, T.Q.V.; validation, H.H., D.T.T.T. and V.V.P.; formal analysis, H.H.; investigation, T.Q.V.; resources, N.T.N.L.; data curation, H.H.; writing—original draft preparation, H.H.; writing—review and editing, H.H.; visualization, V.V.P.; supervision, D.T.T.T.; project administration, T.Q.V.; funding acquisition, T.Q.V. All authors have read and agreed to the published version of the manuscript.

**Funding:** This research received no external funding.

**Institutional Review Board Statement:** All subjects gave their informed consent for inclusion before they participated in the study. The study was conducted in accordance with the Declaration of Helsinki, and the protocol was approved by the Ethics Committee of The Ministry of Home Affairs (MOHA, Vietnamese: Bô Noi vu) (Project identification code: DT.10/22, Host organization: Ha Noi University of Internal Affairs, The Ministry of Home Affairs, Vietnam and date: 17/5/2022).

**Informed Consent Statement:** Not applicable.

**Data Availability Statement:** Not applicable.

**Conflicts of Interest:** The authors declare no conflict of interest.

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
