# Peer review of "Impact of Leadership Behavior on Entrepreneurship in State-Owned Enterprises: Evidence from Civil Servant Management Aimed at Improving Accountability"

_economies, doi:10.3390/economies10100245_

Round 1
Reviewer 1 Report
The manuscript with the title: "Impact of leadership behavior on entrepreneurship of stateowned enterprises: Evidence from civil servants management according to improving accountability" is interesting and well prepared. A few changes need to be made before publishing! I present to the authors my suggestions:
- The abstract needs improvement. The abstract must contain the main aim of the article (research).
- 2. Literature review and methodology is inappropriate due to IMRAD. I recommend changing! 2. Literature review and 3. Materials and methods. Methodology requires fundamental (major) revision and addition of important information! Table 1 needs to be modified... what does the item "Size" mean: 1.96??? Table 1 must contain absolute and relative frequencies!
- I recommend the authors to add research hypotheses that are directly connected to the OLS Model and the OLM Model.
- 3. The results are ok, the authors clearly present the results.
- It is necessary to supplement the Discussion!
- The authors must edit 4. Conclusion, it is necessary to shorten part 4. At the same time, it is necessary to add future research and limitation of the research, which are currently absent.
Overall, the post has a lot of potential, but needs changes. I wish the authors much success in their work.
Author Response
Dear Sir/Madam
We would like to thank you for your interest in the article and for its in-depth suggestions which help us to improve the article in a better way. We have some explanations as follows:
- We improve the abstract
- We change: 2. Literature review and 3. Materials and methods.
- We adjust “size”, Table 1
- We also add research hypotheses that are directly connected to the OLS Model and the Discussion
- We rewrite Conclusion, future research and limitation of the research
These are our explanations for the paper. We are looking forward to receiving your comments to make the paper more and more complete and quality.
With best regard

Reviewer 2 Report
Thank you to the author(s) for examining the Impact of leadership behavior on entrepreneurship of state- owned enterprises and looking at some unique variables that are critical to contemporary management. This study has the potential to be published with some major revisions and editing to make it even stronger. Good luck!
The literature review is thorough, but I will welcome some more recent and significant papers to be taken into consideration, such as:
· Motivating Public Sector Employees: Evidence from Greece”. International Journal of Business and Economic Sciences Applied Research
· A values framework for measuring the influence of ethics and motivation regarding the performance of employees
· The Net Effect of the travel restriction policy on tourism demand: Evidence from Greece
· Strategic Sport Sponsorship Management - A Scale Development and Validation. Journal of Business Research
· Online technologies and sports: A new era for sponsorship
· Cash Holdings, Corporate Performance and Viability of Greek SME: Implications for Stakeholder Relationship Management
The methodology is lacking in terms of the research setting and exact survey methods. I am not clear how the participants were recruited, contacted, surveyed, or incentivized, if any. There needs to be more detail in how the study was designed and who participated.
The section of results was quite clear and well written and presented.
5. Implications for research, practice and/or society: Does the paper identify clearly any implications for research, practice and/or society? Does the paper bridge the gap between theory and practice? How can the research be used in practice (economic and commercial impact), in teaching, to influence public policy, in research (contributing to the body of knowledge)? What is the impact upon society (influencing public attitudes, affecting quality of life)? Are these implications consistent with the findings and conclusions of the paper?:
There was a bit of a reach here in terms of providing conclusions that were not a direct result of the analyses. Some conclusions were also made indicating unique results related to the civil servant management field. These distinctions were not supported or justified by the study. I think the authors need to make sure the conclusions that are presented are based on the individual results from each hypothesis and that overreaches are not made.
6. Quality of Communication: Does the paper clearly express its case, measured against the technical language of the field and the expected knowledge of the journal's readership? Has attention been paid to the clarity of expression and readability, such as sentence structure, jargon use, acronyms, etc.
The paper needs a full edit, to assist the author with some language issues in the writing. There is a lot of superfluous wording, unusual phrasing, which is simply due to the author(s) writing in a second language. This should not diminish their hard work or prevent publication. But a major edit would make it easier to follow for the reader.
Author Response
Dear Sir/Madam
We would like to thank you for your interest in the article and for its in-depth suggestions which help us to improve the article in a better way. We have some explanations as follows:
- We add some some more recent and significant papers to be taken into consideration
- We also add the methodology and explain the paper’s method more
- We rewrite some implications for research and conclusions
- We edit English language and style required
These are our explanations for the paper. We are looking forward to receiving your comments to make the paper more and more complete and quality.
With best regard

Reviewer 3 Report
The paper covers an interesting comparison, but a few things must be better grasped.
The title of the article is unnecessarily broad.
The keywords do not mention that this is research only in Vietnam
The literature review is well conceived but lacks a greater coherence of managers to innovation. I recommend, for example:
10.3390/su9050721
10.1016/j.procs.2019.12.206
10.3390/su10093237
The paper does not have a traditionally conceived methodology section, which can be seen as the main weakness of the paper; section 2.2 does not fully replace it.
The results are clearly and comprehensibly conceived, including a brief description. The statistical methods are adequate.
The paper lacks a discussion section, which would undoubtedly have been helpful.
I recommend the article after completing the Methods and Discussion.
Author Response
Dear sir/madam
Dear Sir/Madam
We would like to thank you for your interest in the article and for its in-depth suggestions which help us to improve the article in a better way. We have some explanations as follows:
- We rewrite the keywords do not mention that this is research only in Vietnam
- We add the review with coherence of managers to innovation.
- We also add 3. Materials and methods and explain the paper’s method more
- We add the Discussion section
These are our explanations for the paper. We are looking forward to receiving your comments to make the paper more and more complete and quality.
With best regard

Round 2
Reviewer 3 Report
The authors have done a great job and the paper can be accepted.